# Viral genome sequence datasets display pervasive evidence of strand-specific substitution biases that are best described using non-reversible nucleotide substitution models

Rita Sianga-Mete[1]*, Penelope Hartnady[1], Wimbai Caroline Mandikumba[1], Kayleigh Rutherford[1], Christopher Brian Currin[2], Florence Phelanyane[3], Sabina Stefan[4], Steven Weaver[5], Sergei L Kosakovsky Pond[5], Darren P Martin[6]

[1]Division of Computational Biology, Institute of Infectious Diseases and Molecular Medicine, Department of Integrative Biomedical Sciences, Faculty of Health Sciences, University of Cape Town, Rondebosch, South Africa; [2]Department of Human Biology, Faculty of Health Sciences, University of Cape Town, Rondebosch, South Africa; [3]Centre for Infectious Disease and Epidemiology Research, School of Public Health and Family Medicine, University of Cape Town, Rondebosch, South Africa; [4]Centre for Biomedical Engineering, School of Engineering, Brown University, Providence, United States; [5]Institute for Genomics and Evolutionary Medicine, Department of Biology, Temple University, Philadelphia, United States; [6]Wellcome Center for Infectious Diseases Research in Africa, Institute of Infectious Disease and Molecular Medicine and Department of Medicine, University of Cape Town, Rondebosch, South Africa

*For correspondence:
rita@aims.ac.za

## eLife Assessment

This **valuable** study revisits the effects of substitution model selection on phylogenetics by comparing reversible and non-reversible DNA substitution models. The authors provide **solid** evidence that (1) it can be beneficial to include non-time-reversible models in addition to general time-reversible models when inferring phylogenetic trees out of simulated viral genome sequence data sets, and that (2) non time-reversible models may fit the real data better than the reversible substitution models commonly used in phylogenetics, a finding consistent with previous work.

**Abstract** Most phylogenetic trees are inferred using time-reversible evolutionary models that assume that the relative rates of substitution for any given pair of nucleotides are the same regardless of the direction of the substitutions. However, there is no reason to assume that the underlying biochemical mutational processes that cause substitutions are similarly symmetrical. We consider two non-reversible nucleotide substitution models: (1) a 6-rate non-reversible model (NREV6) that is applicable to analysing mutational processes in double-stranded genomes, in that complementary substitutions occur at identical rates and (2) a 12-rate non-reversible model (NREV12) that is applicable to analysing mutational processes in single-stranded (ss) genomes, in that all substitution types are free to occur at different rates. Using likelihood ratio and Akaike information criterion-based model tests, we show that, surprisingly, NREV12 provided a significantly better fit than the general time reversible (GTR) and NREV6 models to 21/31 dsRNA and 20/30 dsDNA datasets. As expected,

however, NREV12 provided a significantly better fit to 24/33 ssDNA and 40/47 ssRNA datasets. We tested how non-reversibility impacts the accuracy with which phylogenetic trees are inferred. As simulated degrees of non-reversibility (DNRs) increased, the tree topology inferences using both NREV12 and GTR became more accurate, whereas inferred tree branch lengths became less accurate. We conclude that while non-reversible models should be helpful in the analysis of mutational processes in most virus species, there is no pressing need to use these models for routine phylogenetic inference.

## Introduction

Modelling the nucleotide substitution processes that underlie the diversification of virus genome sequences lies at the heart of many viral evolutionary analyses. The most widely used nucleotide substitution models belong to the general time reversible (GTR) family (*Tavaré, 1986*) and assume that the Markov process of evolution is time-reversible (*Hoff et al., 2016*; *Liò and Goldman, 1998*; *Tavaré, 1986*).

The GTR model is defined by its instantaneous rate matrix $Q_{ij}$ (*Equation 1*), where $Q_{ij}$ defines the instantaneous rate of change from nucleotide $i \in \{A, C, G, T\}$ to nucleotide $j$; subject to the detailed balance condition: $q_{ji}\pi_i = q_{jj}\pi_j$, with rates **q** and equilibrium frequencies **π** (*Squartini and Arndt, 2008*; *Posada, 2003*). The instantaneous rate matrix of the GTR model includes six rate parameters (*a*, *b*, *c*, *d*, *e*, and *f*). Because only products of substitution rates and evolutionary times can be estimated, one of the rate parameters is set to 1 (e.g. *b*), or the entire matrix is normalised to yield one expected substitution per unit time.

$$Q = \{q_{ij}\} = \begin{pmatrix} - & a\pi_C & b\pi_G & d\pi_T \\ a\pi_A & - & c\pi_G & e\pi_T \\ b\pi_A & c\pi_C & - & f\pi_T \\ d\pi_A & e\pi_C & f\pi_G & - \end{pmatrix} \qquad (1)$$

The rate matrix in *Equation 1* is symmetrical, e.g., the relative rate at which A changes to G is the same as the relative rate at which G changes to A.

Time-reversible nucleotide substitution models such as GTR form the basis of almost all nucleotide sequence-focused evolutionary analyses (including those involving eukaryotes, prokaryotes, and viruses) (*Lefort et al., 2017*; *Posada and Crandall, 2001a*; *Posada and Crandall, 2001b*; *Minin et al., 2003*).

The reliability of a phylogenetic tree constructed using a particular nucleotide sequence dataset should be maximised when the evolutionary models used to construct the tree accurately reflect the important aspects of the evolutionary process (*Buckley and Cunningham, 2002*; *Ripplinger and Sullivan, 2008*; *Hoff et al., 2016*). The suitability of different models for describing the evolution of DNA or RNA sequences is, therefore, expected to depend to some degree on the biological and environmental contexts of the sequences being analysed.

Mutations in viral genomes arise due to diverse biotic (such as replication enzyme infidelities, RNA/DNA editing enzymes) and abiotic (such as ionising radiation, inorganic oxidisers, and chemical mutagens) factors (*Sanjuán and Domingo-Calap, 2016*). Mutagenic chemical reactions or types of radiation that, for example, cause G to A or C to U mutations in DNA or RNA are not the same as those that cause A to G or U to C mutations (*Cheng et al., 1992*; *Nguyen et al., 1992*; *Chelico et al., 2006*; *Sharma et al., 2016*). It should not be expected, therefore, that the relative rates of G to A substitution will equal the relative rates of A to G substitution. Instead, in evolving double-stranded (ds) DNA and dsRNA molecules where both strands of the genome are in existence for similar amounts of time, both G to A and C to T substitutions should occur at relatively similar rates. Therefore, for nucleotide sequence datasets derived from any organisms with dsDNA or dsRNA genomes, a non-reversible nucleotide substitution model with a different relative substitution rate category for each of the six possible pairs of complementary nucleotide substitutions (e.g. NREV6 in *Equation 2*), with $q_{AC} = q_{TG}, q_{AG} = q_{TC}, q_{AT} = q_{TA}, q_{CG} = q_{GC}, q_{CT} = q_{GA}, q_{GT} = q_{CA}$, might plausibly provide a better description of mutational processes than GTR (*Baele et al., 2010*; *Wickner, 1993*).

$$Q = \{q_{ij}\} = \begin{pmatrix} - & a\pi_C & b\pi_G & c\pi_T \\ f\pi_A & - & d\pi_G & e\pi_T \\ e\pi_A & d\pi_C & - & f\pi_T \\ c\pi_A & b\pi_C & a\pi_G & - \end{pmatrix} \qquad (2)$$

In the case of ssRNA viruses, ssDNA viruses, retroviruses, and dsRNA/dsDNA viruses where the two complementary genome strands do not exist for equal amounts of time (*Yu et al., 2004*), a model where all 12 different substitutions occur at different rates might be best. Specifically, with ssRNA viruses, ssDNA viruses, and retroviruses, only one of the genome strands (called the virion strand) is packaged into viral particles for transmission and, in many dsRNA viruses, the genome strand that is translated into proteins (called the + strand) exists for longer during the life cycle than does the complementary (or –) strand (*Bruslind, 2020*; *Onwubiko et al., 2020*). In all these viruses, some degree of strand-specific substitution bias is expected to occur (*van der Walt et al., 2008*; *Polak and Arndt, 2008*) such that NREV6 might be anticipated to provide a poorer description of mutational processes than a model such as NREV12 (*Equation 3*), where each of the 12 different types of substitution has a separate rate (*Baele et al., 2010*).

$$Q = \{q_{ij}\} = \begin{pmatrix} - & a\pi_A & b\pi_A & c\pi_A \\ g\pi_C & - & d\pi_C & e\pi_C \\ h\pi_G & i\pi_G & - & f\pi_G \\ j\pi_T & k\pi_T & l\pi_T & - \end{pmatrix} \qquad (3)$$

Because non-reversible models consider the directionality of evolution, they could, in some cases, be used to identify root nodes of phylogenetic trees (*Yap and Speed, 2005*; *Boussau and Gouy, 2006*). It is, however, unclear whether non-reversible models might, in certain situations at least, perform better than reversible models in the context of phylogenetic inference. Although it is possible to use non-reversible nucleotide substitution models such as NREV6 and NREV12 during maximum likelihood-based phylogenetic inference with computer programs such as IQ-TREE (*Nguyen et al., 2015*), these models are not routinely used for phylogenetic inference. This is in part because non-reversible models render several commonly used algorithmic techniques for efficient likelihood computation inapplicable, making inference slower. It is also in part because it remains undetermined whether, under conditions where strand-specific substitution biases are evident, non-reversible models consistently yield substantially more accurate phylogenetic trees than reversible models.

Here, we present evidence that strand-specific nucleotide substitution biases are common within virus genomic sequence datasets such that NREV12 generally provides a significantly better fit than both GTR and NREV6 for such datasets. We then use simulations to demonstrate that whereas strand-specific nucleotide substitution biases reduce the accuracy of phylogenetic inference under both GTR and NREV12, when these biases become extreme, use of NREV12 can yield significantly more accurate phylogenetic trees than GTR.

## Results and discussion
### Non-reversible nucleotide substitution models generally provide a better fit than reversible models to virus sequence datasets

We tested for evidence of non-reversibility of the nucleotide substitution process in 141 virus sequence datasets (33 ssDNA virus datasets, 30 dsDNA virus datasets, 31 dsRNA virus datasets, and 47 ssRNA virus datasets), all consisting of either full genome sequences (for unsegmented viruses) or complete genome component sequences (for viruses with segmented genomes). Specifically, for each dataset, we compared the goodness-of-fit of the GTR+G, NREV6+G, and NREV12+G models (where G represents gamma-distributed nucleotide substitution rates among sites; *Yang, 1994*).

Given that dsDNA viruses such as adenoviruses, papillomaviruses, and herpesviruses have both their DNA strands in existence for similar amounts of time before DNA-dependent-DNA polymerase enzymes copy both their + and – DNA strands during replication (*Hanson, 2009*), we had anticipated that the best fitting substitution model for sequence datasets of these viruses would be NREV6. Using

weighted small sample corrected Akaike information criterion (AIC-c) scores to reveal trends of model support (*Figure 1*), it is surprising that NREV12 was overall the best-supported model (illustrated by the redder hues around the top corner of the dsDNA plot in *Figure 2*). Out of the 30 dsDNA datasets considered, we found that NREV6 provided the best fit to five datasets (HPV18, HPV45, HPV16, BPV, and SV40) and GTR provided the best fit to five (Alphapapillomavirus 6, JC polyomavirus, DPV, RTBV, and DBAV). NREV12 was the best fitting model for the remaining 20 datasets (*Table 1*). Further, likelihood ratio tests (LRTs) revealed strong overall support for NREV12, with this model providing a

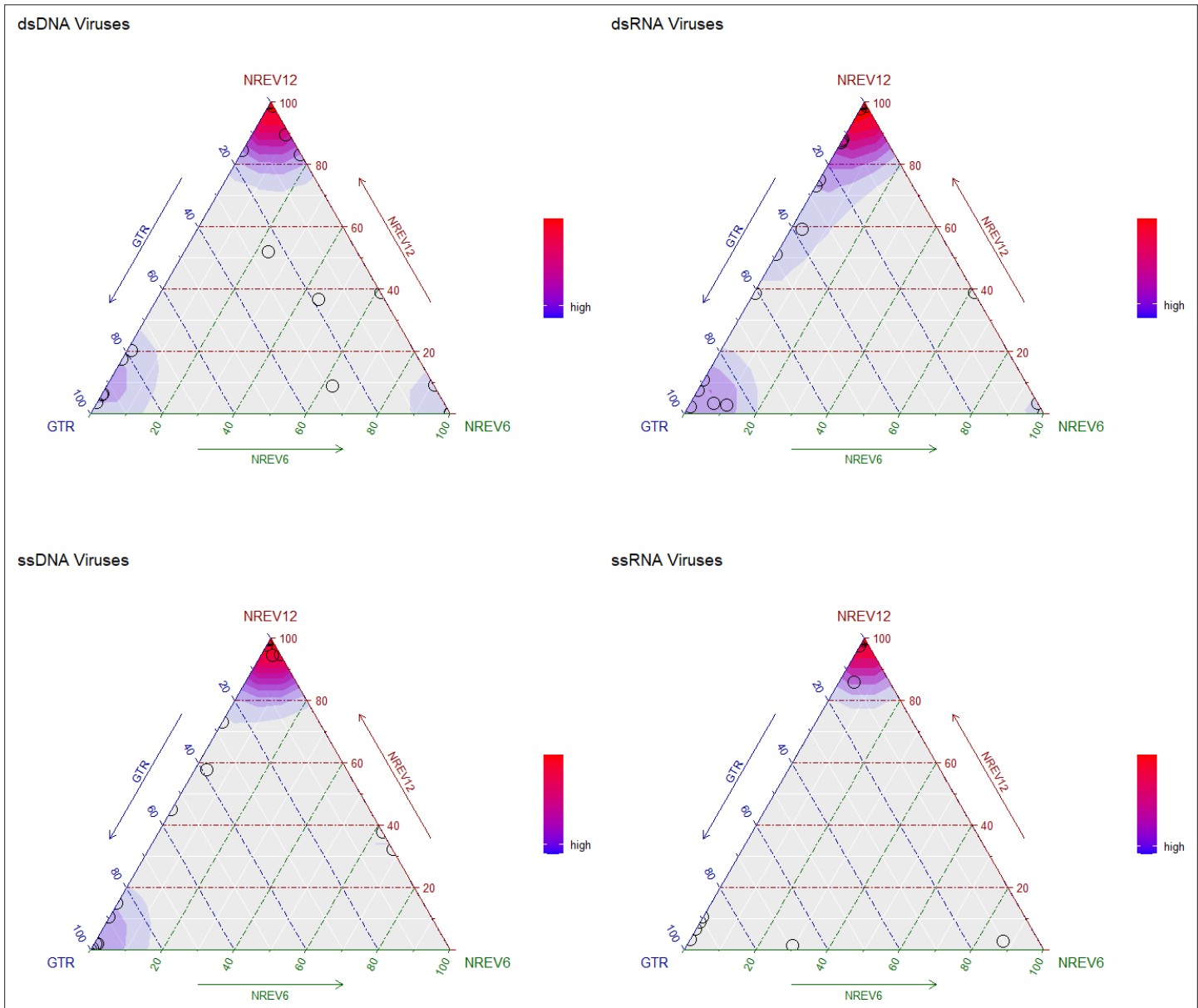

**Figure 1.** Ternary plots illustrating the relative fit of the NREV12, NREV6, and GTR nucleotide substitution models based on weighted small sample corrected Akaike information criterion (AIC-c) scores for 30 dsDNA, 31 dsRNA, 33 ssDNA, and 47 ssRNA virus nucleotide sequence datasets. These plots were produced using the Akaike weights function with an overlaid density function (implemented in the qpcR package of RStudio; *Ritz and Spiess, 2008*) to indicate point densities. Each model is represented by a corner of the triangles, and each circle represents the relative fit of each of the three models to a single nucleotide sequence dataset. The sides of the triangle represent model support axes ranging from 0% to 100%, with the position of a circle in relation to each of the sides of the triangle indicating the probability of models best describing the nucleotide sequence dataset that is represented by that point. Red colours represent a very high density of nucleotide sequence datasets that favour a particular model, blue colours indicate a lower, but still substantial, density of datasets that favour a particular model.

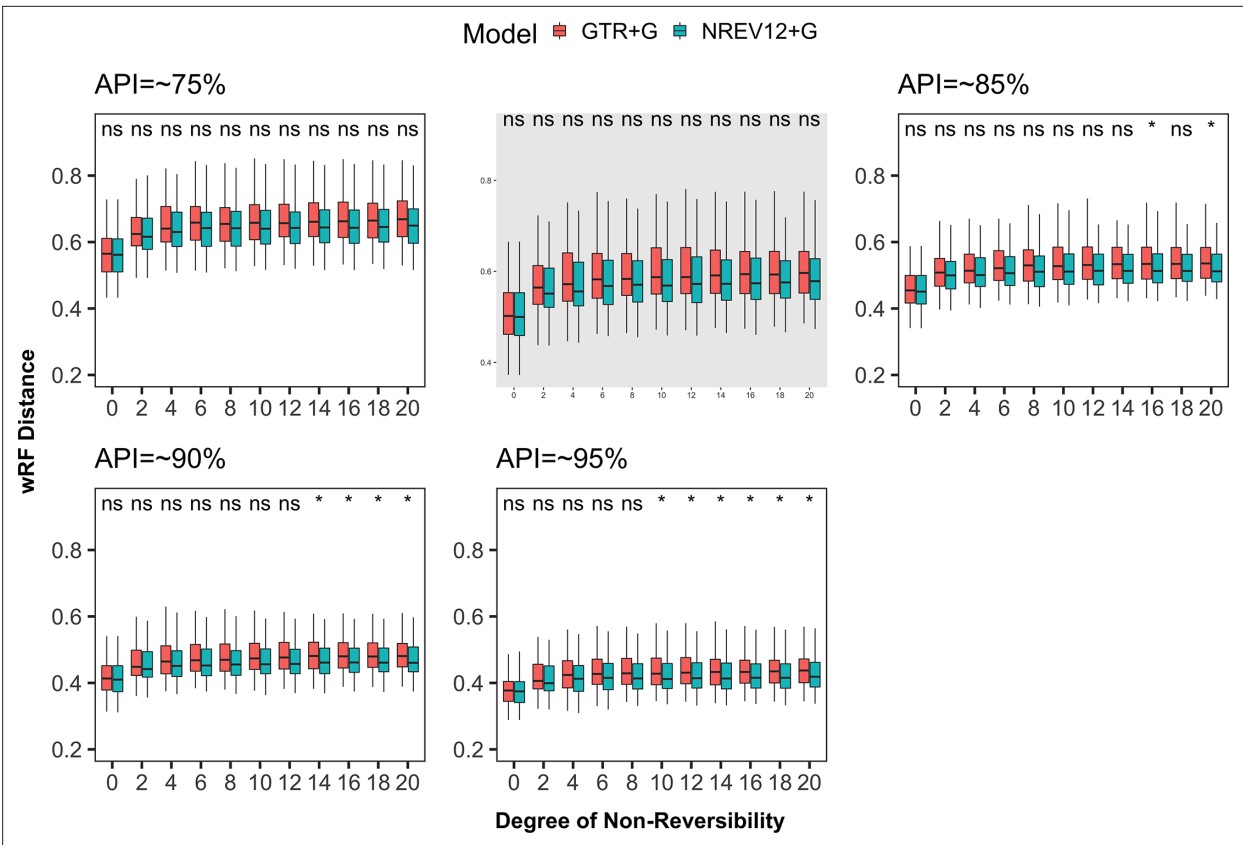

**Figure 2.** Weighted Robinson-Foulds distances between inferred and true phylogenetic trees for datasets simulated with different degrees of nucleotide substitution non-reversibility and different average pairwise sequence identities (APIs) (~75%, ~80%, ~85%, ~90%, and ~95%). 'ns' above a pair of box and whisker plots indicates a paired t-test adjusted p-value of ≥0.05 and '*' indicates a paired t-test adjusted p-value of <0.05.

significantly better fit (p<0.05) than NREV6 for 25/30 of the dsDNA datasets and a significantly better fit than GTR for 24/30 of the datasets.

As NREV6 was not the best fitting model for most of the dsDNA virus datasets, we infer that, in most dsDNA virus species, strand-specific substitution biases are not ignorable. Further, the datasets where NREV6 was not the best fit are from species in families containing other species where NREV6 was the best fit, indicating that such strand-specific substitution biases are unlikely to be a consequence of some broadly conserved feature of viral life cycles in these families (such as ssDNA replicative intermediates). It is instead plausible that these differences may relate to:

i. differences in replication fidelity and/or proofreading efficiency on the leading and trailing DNA strands in some dsDNA virus species (*Grigoriev, 1999*). These differences are common in eukaryotes (*Pavlov et al., 2002*; *Furusawa, 2012*) and prokaryotes (*Fijalkowska et al., 1998*) and, considering that the replication processes of dsDNA viruses analysed here mirror those of their eukaryote hosts, it is perhaps unsurprising that most of these viruses also display some evidence of strand-specific substitution biases.

ii. extra exposure to DNA damage of displaced template strands during unidirectional rolling circle replication in some papillomavirus species such as HPV16 could be a contributor to strand-specific nucleotide substitution biases (*Kusumoto-Matsuo et al., 2011*).

iii. extra time spent by non-coding strands in single-stranded dissociated states during RNA transcription in some papillomavirus and polyomavirus species (*Fernandes and Medeiros Fernandes, 2012*): during transcription processes, the dissociated non-coding strand is transiently more exposed to damage than the coding strand (*Wei et al., 2010*), which might also contribute to strand-specific substitution biases.

Similarly, and equally surprising, we found that NREV12 was overall the best supported model for dsRNA viruses (illustrated by the redder hues around the top corner of the dsRNA plot in *Figure 1*).

**Table 1.** Akaike information criterion (AIC) scores and likelihood ratio test (LRT) results for double-stranded DNA virus datasets. The lowest small sample corrected AIC (AIC-c) scores indicating the best fitting models are in bold.

| Virus family | Dataset | AIC score GTR | AIC score NREV-6 | AIC score NREV-12 | p-Value GTR vs NREV-12 | p-Value NREV-6 vs NREV-12 | DNR |
|---|---|---|---|---|---|---|---|
| | APPV 6 | **35099.5** | 35108.0 | 35102.2 | >0.05 | **0.007** | 0.089 |
| | HPV18_2 | 25202.9 | **25174.6** | 25179.2 | <0.001 | >0.05 | 0.323 |
| | HPV45_2 | 23600.6 | **23599.0** | 23602.9 | >0.05 | >0.05 | 0.285 |
| | HPV16_2 | 29734.0 | **29664.5** | 29665.4 | <0.001 | >0.05 | 0.371 |
| | HPV31 | 24681.4 | 24677.3 | **24672.8** | **0.002** | **0.01** | 0.165 |
| | HPV6_1 | 31199.1 | 31150.0 | **31141.2** | <0.001 | <0.001 | 0.451 |
| | LPV | 67165.7 | 67188.1 | **67145.5** | <0.001 | <0.001 | 0.42 |
| | DPV | **69829.7** | 69889.2 | 69835.1 | >0.05 | <0.001 | 0.056 |
| | XPV | 95455.6 | 95617.1 | **95452.2** | <0.001 | <0.001 | 0.072 |
| Papillomaviridae | BATV | 134821 | 134511 | **133322** | <0.001 | <0.001 | 0.402 |
| | JC_2 | **51806.7** | 51819.6 | 51812.0 | >0.05 | **0.003** | 0.089 |
| | BK_2 | 21472.6 | 21472.7 | **21471.1** | **0.03** | **0.03** | 0.244 |
| | SV40 | 16859.8 | **16858.0** | 16858.4 | **0.037** | >0.05 | 0.567 |
| Polyomaviridae | BPV | 148614.9 | **148573.8** | 148585.2 | <0.001 | >0.05 | 0.064 |
| | CMV | 124083.9 | 124221.0 | **123888.6** | <0.001 | <0.001 | 0.351 |
| | CSSV | 145327.0 | 146575 | **145202** | <0.001 | <0.001 | 0.158 |
| | SVBV | 138575 | 138488.1 | **138464.7** | <0.001 | <0.001 | 0.174 |
| | DBAV | **46495.5** | 46514.1 | 46502.0 | >0.05 | <0.001 | 0.0335 |
| | RTBV | **54987.9** | 55350.1 | 54991 | >0.05 | <0.001 | 0.082 |
| Caulimoviridae | BDV | 376325.2 | 376647.6 | **376029.9** | <0.001 | <0.001 | 0.140 |
| Siphoviridae | CLV | 237362.3 | 237351.8 | **237348.6** | <0.001 | <0.01 | 0.070 |
| Tectiviridae | TTIV | 913864.9 | 913915.4 | **913773.1** | <0.001 | <0.001 | 0.279 |
| | FAV_C | 3074086.7 | 3074207.5 | **3073739.1** | <0.001 | <0.001 | 0.169 |
| | FAV_E | 103482.3 | 103222.7 | **102636.7** | <0.001 | <0.001 | 0.357 |
| | FAV_D | 2326925.6 | 2325719.4 | **2324784.5** | <0.001 | <0.001 | 0.551 |
| | FAV_A | 705328.5 | 705436.5 | **705197.8** | <0.001 | <0.001 | 0.645 |
| | HMAV_B | 103796.7 | 103937.44 | **103753.8** | <0.001 | <0.001 | 10.890 |
| | HMAV_D | 1748635.2 | 1749769 | **1748119.1** | <0.001 | <0.001 | 0.646 |
| | HMAV_C | 2851144.5 | 2851357.1 | **2851133** | **0.006** | <0.001 | 0.0225 |
| Adenoviridae | HMAV_E | 1915044.8 | 1915065.3 | **1914998** | <0.001 | <0.001 | 0.049 |

NREV6 fit only 2 of the 31 dsRNA datasets better than both NREV12 and GTR (Human rotavirus A set H and Fiji virus). NREV12 was found to be the best fitting model for 21/31 datasets and GTR was the best fitting of 8/31 (*Table 2*). In all three Birnaviridae family datasets (which contain virus species with two genome segments) and in 17/22 of Reoviridae family datasets (which contain virus species with 10–12 genome segments), the NREV12 model provided the best fit. Based on the LRTs, strong overall support for NREV12 was found, with this model providing a significantly better fit (p<0.05) in 27/31 dsRNA virus datasets relative to NREV6 and 23/31 datasets relative to GTR.

We anticipated that NREV12 might fit many of these dsRNA datasets better than NREV6 simply because, during their infection cycles, the coding +strand of dsRNA viruses (the one from which

**Table 2.** Akaike information criterion (AIC) scores and likelihood ratio test (LRT) results for double-stranded RNA datasets. The lowest small sample corrected AIC (AIC-c) scores indicating the best fitting models are in bold.

| Virus family | Dataset | AIC score GTR | AIC score NREV-6 | AIC score NREV-12 | GTR vs NREV-12 | NREV-6 vs NREV-12 | DNR |
|---|---|---|---|---|---|---|---|
| | AQBV | 31754.9 | 31853.3 | **31721.9** | <0.001 | <0.001 | 0.219 |
| | GBV_A | 47176.9 | 47347.2 | **47154.8** | <0.001 | <0.001 | 0.142 |
| | IPNV | 79186.2 | 79221.9 | **79182.4** | 0.0145 | <0.001 | 0.162 |
| Birnaviridae | GBV_B | 39313.7 | 39062.8 | **38938.7** | <0.001 | <0.001 | 0.201 |
| | BTV_A | 34803.5 | 34895.1 | **34801.3** | 0.03 | <0.001 | 0.042 |
| | BTV_B | 48849.9 | 48893. | **48837.1** | <0.001 | <0.001 | 0.043 |
| | BTV_C | 28350.9 | 28386.5 | **28350.8** | >0.05 | <0.001 | 0.061 |
| | BTV_D | 24969.1 | 24947.3 | **24894.0** | <0.001 | <0.001 | 0.191 |
| | BTV_F | 20622.7 | 20708.5 | **20610.2** | <0.001 | <0.001 | 0.067 |
| | BTV_G | 63349.9 | 63485.0 | **63345.9** | 0.00426 | <0.001 | 0.040 |
| | BTV_H | 20596.7 | 20685.5 | **20586.1** | <0.001 | <0.001 | 0.118 |
| | BTV_I | 17592.7 | 17622.5 | **17588.8** | 0.01 | <0.001 | 0.095 |
| | BRVA_C | 41206.7 | 41187.4 | **41137.1** | <0.001 | <0.001 | 0.128 |
| | HRVA_A | **17030.5** | 17043.2 | 17035.5 | >0.05 | **0.003** | 0.036 |
| | HRVA_B | **8275.1** | 8280.3 | 8281.7 | >0.05 | >0.05 | 0.087 |
| | HRVA_C | 12815.1 | 12842.6 | **12807.6** | 0.003 | <0.001 | 0.132 |
| | HRVA_D2 | **8036.8** | 8041.0 | 8043.7 | >0.05 | >0.05 | 0.057 |
| | HRVA_E | **7045.9** | 7056.1 | 7053.3 | >0.05 | **0.02** | 0.102 |
| | HRVA_F | **7046.0** | 7056.7 | 7053.4 | >0.05 | **0.02** | 0.0710 |
| | HRVA_G | 18424.2 | 18434.0 | 18425.1 | >0.05 | **<0.001** | 0.123 |
| | HRVA_H | 20431.4 | **20413.87** | 20420.5.6 | 0.002 | >0.05 | 0.163 |
| | PRVA_A | 28540.7 | 28441.9 | **28398.7** | <0.001 | <0.001 | 0.204 |
| | PRVA_B | 14757.7 | 14775.5 | **14732.6** | <0.001 | <0.001 | 0.351 |
| | HRVC_A | 6713.2 | 6718.2 | **6712.3** | 0.045 | 0.007 | 0.124 |
| | PTOV | 202011.3 | 202106.5 | **201878.5** | <0.001 | <0.001 | 0.039 |
| Reoviridae | FJV_B | 9274.1 | **9250.0** | 9250.9 | <0.001 | >0.05 | 0.194 |
| | EDV | 1771992.8 | 1772689.1 | **1771950.6** | <0.001 | <0.001 | 0.121 |
| Endornaviridae | BPAV | **70386.5** | 70540.2 | 70390.7 | >0.05 | **0.00** | 0.047 |
| | TTV | 617302.6 | 617462.6 | **617172.9** | <0.001 | <0.001 | 0.052 |
| Totiviridae | GDV | **80435.8** | 80396.5 | 80387.7 | <0.001 | 0.002 | 0.109 |
| Hypoviridae | HPV | 66859.8 | 66899.8 | **66857.8** | 0.03 | <0.001 | 0.057 |

protein translation occurs) tends to exist for longer periods within an infected cell than the non-coding –strand. Specifically, there are two main steps during double-stranded RNA virus replication (*Wickner, 1993*). Firstly, synthesis of the viral +strands from a dsRNA template occurs in the cytoplasm within viral particles. These +strands exist within the cell for prolonged periods in the absence of complementary –strands and are used as templates for translation of viral proteins. In the second step, the +strands remaining after translation act as templates for –strand synthesis, resulting in the formation of new dsRNA molecules. The +strands of dsRNA viruses are therefore likely more impacted by

mutational processes, which in turn could explain the pervasive strand-specific substitution biases seen in this group of viruses.

For the ssRNA and ssDNA viruses where one genome strand exists during the virus life cycle for far longer periods of time than the other such that complementary substitutions would not be expected to occur at similar rates, we anticipated that NREV12 should provide a better fit than both NREV6

**Table 3.** Small sample corrected Akaike information criterion (AIC-c) scores and likelihood ratio test (LRT) results for single-stranded DNA datasets.

The lowest AIC-c scores indicating the best fitting models are in bold.

| Virus family | Dataset | AIC score GTR | AIC score NREV-6 | AIC score NREV-12 | p-Value GTR vs NREV-12 | p-Value NREV-6 vs NREV-12 | DNR |
|---|---|---|---|---|---|---|---|
| | BBTV M | 15044.3 | 15207.9 | **14984.4** | <0.001 | <0.001 | 0.662 |
| | BBTV N | 10605.6 | 10686.2 | **10595.2** | <0.001 | <0.001 | 0.533 |
| | BBTV R | 18484.5 | 18544 | **18480.8** | >0.05 | <0.001 | 0.609 |
| | BBTV S | 12718.9 | 12757.2 | **12707.3** | <0.001 | <0.001 | 0.728 |
| | CCDV | **38622.7** | 38632.0 | 38630.5 | >0.05 | 0.03 | 0.050 |
| | MDV | 36232.8 | **36063** | 36064 | <0.001 | >0.05 | 0.142 |
| | PYDV | 56138.4 | 56076.6 | **56056.4** | <0.001 | <0.001 | 0.187 |
| Nanoviridae | FBNS | 100153.6 | 100135.6 | **100120.5** | <0.001 | <0.001 | 0.098 |
| | Begomo 5 | **28192.1** | 28311.9 | 28192.5 | >0.05 | <0.001 | 0.1995 |
| | Begomo 6 | 16743.0 | **16722.6** | 16724.1 | <0.001 | >0.05 | 0.214 |
| | Begomo 9 | 8517.6 | 8540.8 | **8515.6** | 0.03 | <0.001 | 0.312 |
| | Dicot 1 | 44730.7 | 44594.3 | **44583.3** | <0.001 | <0.001 | 0.200 |
| | Dicot 2 | **39909.9** | 39919.8 | 39917.9 | >0.05 | <0.001 | 0.100 |
| | MSV | **252645.3** | 254347.5 | 254347.5 | <0.001 | <0.001 | 0.144 |
| | PanSV | 94601.2 | 94600.3 | **94593.7** | <0.001 | <0.001 | 0.182 |
| Geminiviridae | WDV | 35301.7 | 35313.2 | **35253.8** | <0.001 | <0.001 | 0.1033 |
| | BFDV | 17256.7 | 17262.7 | **17246.7** | <0.001 | <0.001 | 0.224 |
| | DG_CV | **12754.8** | 12779.5 | 12758.3 | >0.05 | <0.001 | 0.116 |
| | PiCV | **19180.5** | 19192.5 | 19191.0 | >0.05 | 0.04 | 0.117 |
| | CCCC | 84435.7 | 84377.4 | **84315.3** | <0.001 | <0.001 | 0.132 |
| | BTC | 262910.4 | 262060.1 | **261985.4** | <0.001 | <0.001 | 0.178 |
| | POCV2 | 24940.9 | 24953.8 | **24915.8** | <0.001 | <0.001 | 0.162 |
| Circoviridae | CCV | 90307.9 | 90301.5 | **90285.9** | <0.001 | <0.001 | 0.114 |
| | TTV_1 | 825811 | 826800 | **825292** | <0.001 | <0.001 | 0.513 |
| Anelloviridae | TTSV | 332287.9 | 332397.4 | **332258.2** | <0.001 | <0.001 | 1.560 |
| | MVM | 26756.3 | 26743.9 | **26686.9** | <0.001 | <0.001 | 0.148 |
| | HPV | 67051.2 | 67080.1 | **67001.8** | <0.001 | <0.001 | 0.235 |
| | CPV | 85731 | 85695 | **85689.3** | <0.001 | 0.007 | 0.062 |
| | PPV | 163006.8 | 163090.7 | **162995.9** | <0.001 | <0.001 | 0.143 |
| Parvoviridae | CAV_P | 37073.3 | 37115.5 | **37065.7** | <0.001 | <0.001 | 0.162 |
| Microviridae | BMV | 31175.3 | 31164.8 | **31147.3** | <0.001 | <0.001 | 0.188 |
| | APV | 85700.2 | 85617.4 | **85402.8** | <0.001 | <0.001 | 0.204 |
| Pleolipoviridae | BPV | 204797.5 | 204802.3 | **204796.7** | 0.04 | 0.007 | 0.064 |

and GTR. Indeed, for ssRNA viruses, NREV12 was a better fit than NREV6 and GTR for 40/47 of the ssRNA datasets and 24/33 of the ssDNA virus datasets (*Figure 1*). Of the nine ssDNA virus datasets where NREV12 was not the best fitting model, GTR fit 7/9 better and NREV6 fit 2/9 better (*Table 3*). Of the seven ssRNA datasets where NREV12 was not the best fitting model, GTR fit 6/7 better and NREV6 fit 1/7 better.

Based on the LRTs, strong overall support for NREV12 was found with this model providing a significantly better fit (p<0.05) than NREV6 for 45/47 of the ssRNA virus datasets (*Table 4*) and 31/33 of the ssDNA virus datasets (*Table 3*). Similarly, based on LRTs, NREV12 provided a significantly better fit than GTR for 27/33 of the ssDNA virus datasets (*Table 3*) and 40/47 of the ssRNA virus datasets (*Table 4*).

We found that the degree of non-reversibility (DNR) estimates alone did not cleanly differentiate between datasets for which NREV12 was or was not best supported (*Tables 1–4*). For the 107 nucleotide sequence datasets with a model preference of NREV12, 10 had estimated DNRs that were greater than 0.5, 13 had DNRs between 0.25 and 0.5, and 84 had DNRs between 0.0225 and 0.25. For the 10 nucleotide sequence datasets with a model preference of NREV6, one had an estimated DNR greater than 0.5, four had estimated DNRs between 0.25 and 0.5, and five had estimated DNRs between 0.064 and 0.25 (*Figure 1*). For the 24 nucleotide sequence datasets with a model preference of GTR, none had estimated DNRs greater than 0.5, one had an estimated DNR between 0.25 and 0.5, and the remainder had estimated DNRs between 0.0335 and 0.25.

The dsDNA virus dataset with the highest DNR was Human mastadenovirus D (DNR = 0.646), the dsRNA virus dataset with the highest estimated DNR was Porcine_rotavirus_B (0.351), the ssRNA virus dataset with the highest DNR was SARS-CoV-2 (DNR = 1.536), and the ssDNA virus dataset with the highest DNR was Torque teno sus virus (DNR = 1.56).

Therefore, while NREV12 appears to be generally more appropriate than either NREV6 or GTR for describing mutational processes in ssRNA, ssDNA, dsDNA, and dsRNA viruses, this might only be particularly relevant from a practical perspective when datasets of these viruses yield DNR estimates that are greater than 0.25. For such datasets, NREV12 (and possibly NREV 6 in some instances) might be especially useful for both determining the direction of evolution across phylogenetic trees (i.e. it could potentially be used to root these trees) and for quantifying genomic strand-specific nucleotide substitution biases (*Harkins et al., 2009*).

## Assessing the impacts of model misspecification on phylogenetic tree inference

To determine whether it is worthwhile to use NREV12 rather than GTR for phylogenetic inference when NREV12 is the best fitting nucleotide substitution model, we used simulated datasets to compare the accuracy of phylogenetic trees inferred using these models.

We found that, regardless of dataset diversity and the nucleotide substitution model used, phylogenetic inference tended to become less accurate (i.e. weighted Robinson-Foulds [wRF] scores increased) as DNR increased (*Figure 2*). This tendency was, however, more pronounced when using a (mis-specified) GTR model than when using a (correctly specified) NREV12 model with, for any given dataset having DNR>0, the use of NREV12 tending to yield more accurate phylogenetic trees than when GTR was used. There were, however, only statistically significant improvements (p<0.05, paired t-test) in the accuracy of phylogenetic trees inferred using NREV12 relative to those inferred using GTR in lower diversity datasets (i.e. those with average pairwise nucleotide sequence identities (APIs) of 85%, 90%, and 95%) and then only for DNR>8. It is noteworthy that the highest estimated DNR in any of the empirical datasets that we analysed was 1.56 more than fourfold lower than the point where statistically significant differences in phylogenetic inference accuracies became apparent in the simulated datasets.

# Materials and methods
## Virus sequence datasets and phylogenetic trees

We obtained viral nucleotide sequences from the National Centre for Biotechnology Information Taxonomy database (http://www.ncbi.nlm.nih.gov/taxonomy) and the Los Alamos National Laboratory HIV sequence database (https://www.hiv.lanl.gov/content/index). These included gene and

**Table 4.** Small sample corrected Akaike information criterion (AIC-c) scores and likelihood ratio test (LRT) results for single-stranded RNA datasets.

The lowest AIC-c scores indicating the best fitting models are in bold.

| Virus family | Dataset | AIC score GTR | AIC score NREV-6 | AIC score NREV-12 | p-Value GTR vs NREV-12 | p-Value NREV-6 vs NREV-12 | DNR |
|---|---|---|---|---|---|---|---|
| | HAV | 94580.7 | 94926.3 | **94548.1** | <0.001 | <0.001 | 0.096 |
| | BAV | 188307.1 | 188572.9 | **188144.9** | <0.001 | <0.001 | 0.108 |
| | MMV | 281072.2 | 281094.5 | **281076.9** | >0.05 | <0.001 | 0.072 |
| | PAV | 150626.88 | 150827.6 | **150609.5** | <0.001 | <0.001 | 0.069 |
| | CKV | 90902.3 | 91233.1 | **90873.0** | <0.001 | <0.001 | 0.083 |
| | GA | 64998.5 | 65223.9 | **64975.9** | <0.001 | <0.001 | 0.110 |
| Astroviridae | CAV_A | 85558.8 | 85617.4 | **85547.3** | <0.01 | <0.001 | 0.076 |
| | CMV RNA1 | 34197.5 | 34198.8 | **34147.7** | <0.001 | <0.001 | 0.124 |
| | CMV RNA2 | **31398.2** | 31455.9 | 31388.7 | <0.001 | <0.001 | 0.091 |
| | CMV RNA3 | **24337.2** | 24360.3 | 24343.9 | >0.05 | <0.001 | 0.073 |
| | AMS | **24337.2** | 24360.3 | 24343.9 | >0.05 | <0.001 | 0.073 |
| Bromoviridae | PSV | 67707 | 67786.5 | **67691** | <0.001 | <0.001 | 0.048 |
| | LAV | 73042.8 | 73102.4 | **72984.6** | <0.001 | <0.001 | 0.120 |
| | NoV | 207667.2 | 207777.5 | **207660** | <0.001 | <0.001 | 0.047 |
| Caliciviridae | VSV | 235936.4 | 236051.4 | **235913.3** | <0.001 | <0.001 | 0.046 |
| Closteroviridae | CTV | **30062.2** | 29980.4 | 29960.1 | <0.001 | <0.001 | 0.272 |
| | DGV_T1 | 69771.9 | 70030.5 | **69776.2** | >0.05 | <0.001 | 0.063 |
| Flaviviridae | JEV | 146920.8 | 148101.5 | **146885.5** | <0.001 | <0.001 | 0.091 |
| | HPVE1 | 200439.5 | 200863.8 | **200179.8** | <0.001 | <0.001 | 0.073 |
| Hepeviridae | HPVE2 | 155709.1 | 155983.8 | **155518.6** | <0.001 | <0.001 | 0.088 |
| | ENV_A | 552287.9 | 553535.5 | **551794.1** | <0.001 | <0.001 | 0.061 |
| | HRV_A | 102218.7 | 102267.0 | **101550.7** | <0.001 | <0.001 | 0.285 |
| | AIV | 101073.1 | 101136.7 | **101052.2** | <0.001 | <0.001 | 0.093 |
| | AHP | 139635.7 | 140119.6 | **139506.9** | <0.001 | <0.001 | 0.170 |
| | ECV | 82078.9 | 82181.0 | **82065.8** | <0.001 | <0.001 | 0.066 |
| | CDV | 130551.3 | 130896.7 | **130478.3** | <0.001 | <0.001 | 0.086 |
| | TCV | 53027.3 | 53029 | **53023** | 0.0151 | 0.0422 | 0.033 |
| Picornaviridae | FMDV | 455180.6 | 455582.6 | **454806.1** | <0.001 | <0.001 | 0.117 |
| Fusariviridae | FRV | **52413.1** | 52470.6 | 52418.4 | >0.05 | <0.001 | 0.076 |

*Table 4 continued on next page*

*Table 4 continued*

| Virus family | Dataset | AIC score GTR | AIC score NREV-6 | AIC score NREV-12 | p-Value GTR vs NREV-12 | p-Value NREV-6 vs NREV-12 | DNR |
|---|---|---|---|---|---|---|---|
| | HIV1_setA | 344014.4 | 344295.1 | **343669.7** | <0.001 | <0.001 | 0.237 |
| | HIV1_M | 80764.1 | 80829.5 | **80668.1** | <0.001 | <0.001 | 0.442 |
| | HIV1_setC | 180575.0 | 180702.3 | **180494.4** | <0.001 | <0.001 | 0.107 |
| | HIV1_setD | 298489.9 | 298695.3 | **298260.6** | <0.001 | <0.001 | 0.133 |
| | HIV1_setE | 289111.3 | 289292.1 | **288941.9** | <0.001 | <0.001 | 0.112 |
| | HIV1_setF | 214375.9 | 214692.2 | **214289.4** | <0.001 | <0.001 | 0.148 |
| | EIV | 126149 | 126365.4 | **125300** | <0.001 | <0.001 | 0.192 |
| | BIV | **24505.2** | 24506.9 | 24513.2 | >0.05 | >0.05 | 0.15 |
| | FIV | 164542.1 | 164487.9 | **164260.4** | <0.001 | <0.001 | 0.114 |
| | CAV | 351329.9 | 351871.5 | **350721.9** | <0.001 | <0.001 | 0.174 |
| Retroviridae | SIV | 110731.2 | 110816 | **110663.3** | <0.001 | <0.001 | 0.144 |
| Filoviridae | Ebola_2 | 53147.3 | **53143.0** | 53149.9 | >0.05 | >0.50 | 0.264 |
| | Flu A 2 | 82872.8 | 83010.2 | **82849.7** | <0.001 | <0.001 | 0.27 |
| Orthomyxo-viridae | Flu B 1 | 50090.4 | 50144.1 | **50060.9** | <0.001 | <0.001 | 0.311 |
| | SARS-COV1 | 214715.3 | 214968.5 | **214644.39** | <0.001 | <0.001 | 0.198 |
| | SARS-COV2 | 15715.4.2 | 15715.6 | **15696.7** | <0.001 | <0.001 | 1.536 |
| | SARB | 573966.3 | 573815.1 | **572517.0** | <0.001 | <0.001 | 0.301 |
| Coronaviridae | MERS-COV | 516683.2 | 516983.4 | **516608.9** | <0.001 | <0.001 | 0.169 |

whole-genome sequences for viruses with ssRNA, ssDNA, dsRNA, and dsDNA genomes (datasets are summarised in *Supplementary file 1*). An outgroup sequence from a closely related virus species was added to each dataset to help root phylogenetic trees. The sequences in each of the datasets were aligned using MUSCLE (*Edgar, 2004*) implemented in Aliview (*Larsson, 2014*), and maximum likelihood phylogenetic trees were constructed from each alignment using RAxML v8.2 (*Stamatakis, 2016*).

## Model testing

We evaluated the fit of NREV12, NREV6, and GTR to the 141 individual sequence datasets using a previously published model test (*Harkins et al., 2009*) implemented as a HyPhy package module (*Pond and Muse, 2005*). This script (https://github.com/veg/hyphy-analyses/tree/master/Nucleotide-NonREV; *Delport and Kosakovsky Pond, 2023*) is also available as a module in the Datamonkey web server (*Weaver et al., 2018*), which takes as input a rooted maximum likelihood phylogenetic tree (minus the rooting sequence) and its corresponding nucleotide sequence alignment. The three models described above: GTR, NREV6, and NREV12 are then fitted to the data using maximum likelihood (ML). The equilibrium frequencies (EF) of the GTR model match those empirically observed in the alignment, while EFs for NREV6 and NREV12 are inferred by the model, satisfying the condition $\pi Q = 0$. An additional model NREV12 + F is estimated, where the distribution of nucleotides at the root of the tree is estimated by maximum likelihood, instead of being set to the empirical frequencies. Nested models were compared by the LRT with significance evaluated using the $\chi_d^2$ distribution with $d$=difference in degrees of freedom. For all models, we also computed the small AIC-c score.

## Quantification of non-reversibility

We further defined the DNR as the absolute difference between the relative rate differences of two nucleotide pairs; i.e., for two nucleotides, *x* and *y*, there exists a relative rate of *x* to *y* substitutions that we will refer to as *m*, and a relative rate of *y* to *x* substitutions that we will refer to as *n*. Under the

NREV12 model, the DNR between *x* and *y* is defined simply as the absolute difference between *m* and *n*: ($|m\text{-}n|$) and will hereby be referred to as $ij_{DNR}$, where *i* and *j* are two nucleotides. We use DNR as a mathematical representation of the degree to which the rates of all pairs of reverse substitutions differ from one another. For each of the 140 individual viral alignments, we calculated the average DNR of the six $ij_{DNR}$ estimates inferred using the NREV12 model.

## Simulations for testing the impact of non-reversible evolution on phylogenetic inference

We tested the accuracy of phylogenetic tree inference under reversible and non-reversible models using simulated datasets with varying APIs evolved under the NREV12 model with different DNRs. The goal of these tests was not to exhaustively evaluate model misspecification issues during phylogenetic tree inference, but rather to check, in instances where viral taxa are known to be evolving in a detectably non-reversible manner (i.e. where NREV12 or NREV6 fits the data better than GTR), whether not accounting for this might decrease the accuracy of phylogenetic inference. Using IQ-TREE, a phylogenetic inference program that has the option to apply an NREV12-like model (referred to in IQ-TREE as the UNREST model), a phylogenetic tree was inferred from an alignment of real sequences (Avian Leukosis virus) (*Figure 3*) with an API of ~90%. The branch lengths on this tree were then scaled to create four other phylogenetic trees representing sequences with approximately 95%, 85%, 80%, and 75% API. These five trees are hereafter referred to as 'true' trees, and each individual tree was used as the starting point of a different set of simulations.

Phylogenetic trees were inferred from these 5500 simulated datasets and compared to the phylogenetic trees used to simulate the datasets (i.e. the true trees) using wRF distances to assess the impact of varying DNR on the accuracy of phylogenetic inference. We further tested whether the accuracy of phylogenetic inference could be improved for sequences that had evolved under DNR >0

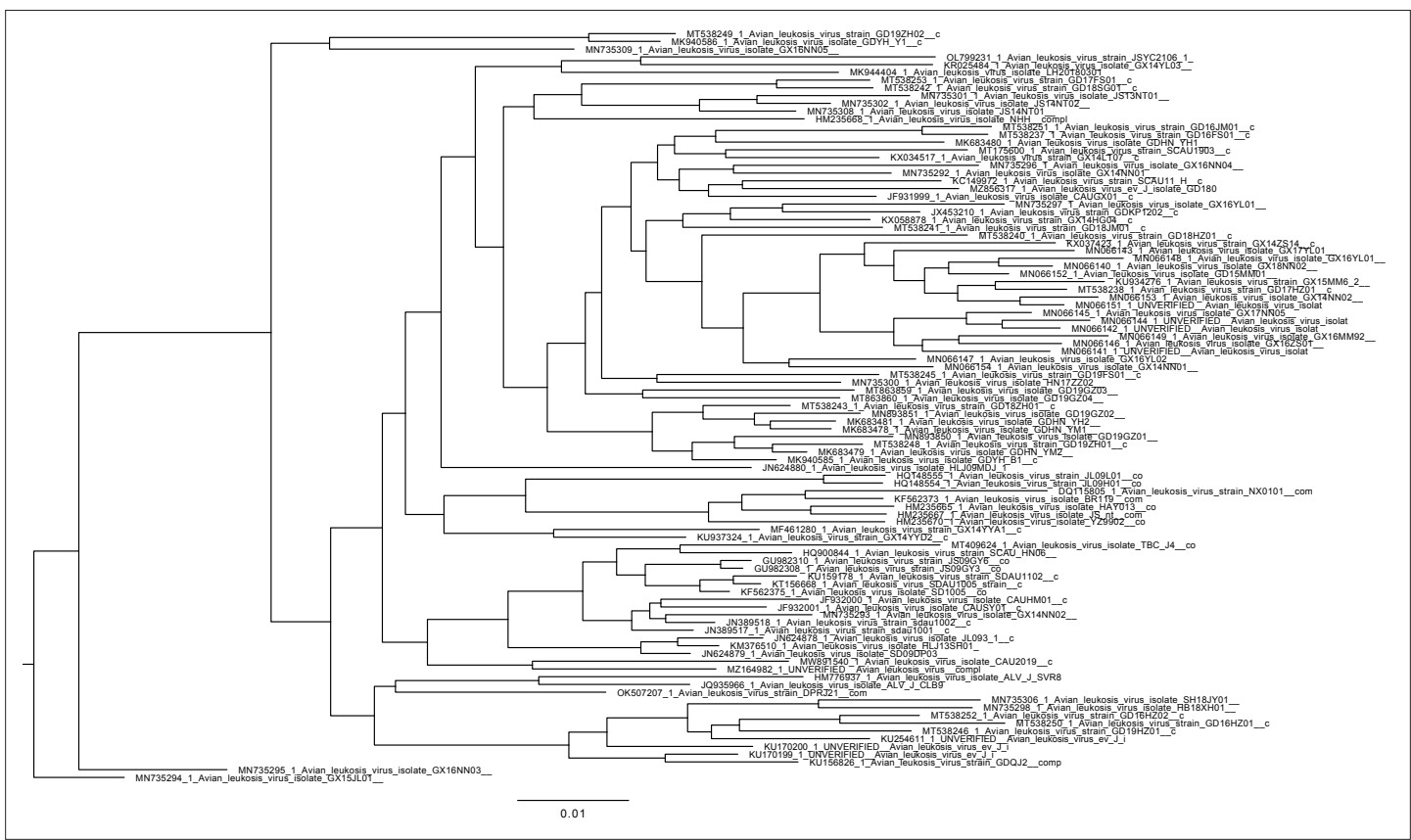

**Figure 3.** Phylogenetic tree inferred from an alignment of real sequences (Avian Leukosis virus) that was used to simulate datasets with degrees of non-reversibility (DNRs) varying from 0 to 20. The alignment of Avian Leukosis virus had an average sequence identity (API) of ~90%, and the branches of this tree were scaled to produce four other trees reflecting branch tip sequences with approximate pairwise identities of ~75%, ~80%, ~85%, and~95%.

by using NREV12 instead of GTR. Specifically, for every simulated dataset, a phylogenetic tree was inferred using GTR, and another using NREV12 and the wRF distances of each of these trees to the true tree was determined. For each of the analysed DNRs, a paired t-test was then used to compare the wRF scores of trees inferred using GTR and NREV12. We were particularly interested in determining whether trees inferred using a mis-specified model (i.e. GTR in this case) would be less accurate than trees inferred with a correctly specified model (i.e. NREV12).

To test whether failure to account for non-reversibility might decrease the accuracy of phylogenetic inference, we simulated the evolution of 5500 nucleotide sequence alignments evolved non-reversibly under varying DNR along the five true phylogenetic trees: 100 datasets per true tree per simulated DNR. Specifically, simulations were done using HyPhy (**Pond and Muse, 2005**), with relative rates ranging from a completely reversible matrix (**Equation 4**)

$$Q = \{q_{ij}\} = \begin{pmatrix} - & 0.166 & 1 & 0.14 \\ 0.166 & - & 0.131 & 1.101 \\ 1 & 0.131 & - & 0.188 \\ 0.14 & 1.101 & 0.188 & - \end{pmatrix}$$

(4)

representing DNR = 0 – through matrices with DNR = 2, 4, 6, 8, 10, 12, 14, 16, 18, and 20 (**Table 5**). These baselines-simulated substitution rates are reflective of those seen in empirical viral nucleotide sequence datasets.

At each DNR, the relative rates used conformed to standard measures of non-reversibility under the Kolmogorov conditions according to which non-reversibly evolving sequence datasets should yield three irreversibility indices (IRI1, IRI2, and IRI3) that are all non-zero (**Squartini and Arndt, 2008**). It should be noted that all simulations under NREV12 were performed under the stationarity criterion: $\pi e^{Qt} = \pi$ (where $Q$ is the rate matrix, $\pi$ is the nucleotide frequency distribution, and $t \geq 0$).

## Quantifying the accuracy of phylogenetic inferences

We used the wRF (implemented in the R phangorn package; **Schliep, 2011**) to quantify differences between the true trees used to simulate datasets and the trees inferred from these datasets using the GTR or NREV12 models. wRF considers differences between both the topology and branch lengths of actual and inferred trees (**Kuhner and Yamato, 2015; Robinson and Foulds, 1981**).

**Table 5.** Relative rate change for C to A, G to A, A to T, G to C, T to G, and C to T mutations under the 11 degrees of non-reversibility alongside the maintained rates for A to C, A to G, T to A, C to G, G to T, and T to C.

| Degree of non-reversibility (DNR) | Relative rates of different nucleotide substitution types (from-to) | | | | | | | | | | | |
|---|---|---|---|---|---|---|---|---|---|---|---|---|
| | C-A | A-C | G-A | A-G | A-T | T-A | G-C | C-G | T-G | G-T | C-T | T-C |
| 0 | 0.166 | 0.166 | 1 | 1 | 0.14 | 0.14 | 0.131 | 0.131 | 0.118 | 0.118 | 1.101 | 1.101 |
| 2 | 2.166 | 0.166 | 3 | 1 | 2.14 | 0.14 | 2.131 | 0.131 | 2.118 | 0.118 | 3.101 | 1.101 |
| 4 | 4.166 | 0.166 | 5 | 1 | 4.14 | 0.14 | 4.131 | 0.131 | 4.118 | 0.118 | 5.101 | 1.101 |
| 6 | 6.166 | 0.166 | 7 | 1 | 6.14 | 0.14 | 6.131 | 0.131 | 6.118 | 0.118 | 7.101 | 1.101 |
| 8 | 8.166 | 0.166 | 9 | 1 | 8.14 | 0.14 | 8.131 | 0.131 | 8.118 | 0.118 | 9.101 | 1.101 |
| 10 | 10.166 | 0.166 | 11 | 1 | 10.14 | 0.14 | 10.131 | 0.131 | 10.118 | 0.118 | 11.101 | 1.101 |
| 12 | 12.166 | 0.166 | 13 | 1 | 12.14 | 0.14 | 12.131 | 0.131 | 12.118 | 0.118 | 13.101 | 1.101 |
| 14 | 14.166 | 0.166 | 15 | 1 | 14.14 | 0.14 | 14.131 | 0.131 | 14.118 | 0.118 | 15.101 | 1.101 |
| 16 | 16.166 | 0.166 | 17 | 1 | 16.14 | 0.14 | 16.131 | 0.131 | 16.118 | 0.118 | 17.101 | 1.101 |
| 18 | 18.166 | 0.166 | 19 | 1 | 18.14 | 0.14 | 18.131 | 0.131 | 18.118 | 0.118 | 19.101 | 1.101 |
| 20 | 20.166 | 0.166 | 21 | 1 | 20.14 | 0.14 | 20.131 | 0.131 | 20.118 | 0.118 | 21.101 | 1.101 |

## Conclusion

The non-reversible nucleotide substitution model, NREV12, provides a substantially better fit to most virus nucleotide sequence datasets than does the widely used reversible substitution model, GTR. NREV12 also provides a better fit to most virus nucleotide sequence datasets than does NREV6; a non-reversible model that would be expected to best describe the evolution of double-stranded genome sequences that display no strand-specific nucleotide substitution biases. This suggests that, contrary to our expectations, substantial strand-specific nucleotide substitution biases (i.e. estimated DNRs>0.25) are common during viral evolution irrespective of genome type. Such biases should be expected for any viruses where one genome strand either is in existence for substantially longer periods of time than the other, or is more exposed to mutagenic processes than the other during transmission, replication, or gene expression.

We had anticipated that, given evidence of sequences evolving both non-reversibly and with strand-specific substitution biases, inferring trees using a model such as NREV12 that appropriately accounts for this might: (1) minimise the impact of increasing DNR on the accuracy of phylogenetic inference (i.e. wRF scores presented in blue in *Figure 2* might have been expected to not increase with increasing DNR) and (2) yield significantly more accurate phylogenetic inferences than when using GTR for all datasets where NREV12 was the most appropriate model and DNRs were greater than zero. However, increasing DNR clearly decreased the accuracy of phylogenetic inference even when using NREV12, and, for datasets where DNRs were greater than zero, using GTR did not consistently yield significantly less accurate phylogenetic inferences than those attained using NREV12. From a practical perspective, choosing a non-reversible nucleotide substitution model to construct phylogenetic trees from virus genome sequences that display strand-specific nucleotide substitution biases is not guaranteed to yield more accurate phylogenetic trees. Nevertheless, in instances where strand-specific substitution biases are higher than ~0.5 (such as are found in our SARS-CoV-2, Torque teno sus virus, and Banana bunchy top virus datasets), it may be prudent to select a model such as NREV12 (such as is implemented in programs like IQ-TREE) over GTR as the better of two suboptimal choices.

The lack of available data regarding the proportions of viral life cycles during which genomes exist in single- and double-stranded states makes it difficult to rationally predict the situations where the use of models such as GTR, NREV6, and NREV12 might be most justified: particularly in light of the poor overall performance of NREV6 and GTR relative to NREV12 with respect to describing mutational processes in viral genome sequence datasets. We therefore recommend case-by-case assessments of NREV12 vs NREV6 vs GTR model fit when deciding whether it is appropriate to consider the application of non-reversible models for phylogenetic inference and/or phylogenetic model-based analyses such as those intended to test for evidence of natural selection or the existence of molecular clocks.

## Declarations

### Ethics

The University of Cape Town ethics committee declared that this research did not need ethics approval due to the use of freely accessible nucleotide sequences obtained from the National Centre for Biotechnology Information Taxonomy database (https://www.ncbi.nlm.nih.gov/taxonomy) and the Los Alamos National Laboratory HIV sequence database (https://www.hiv.lanl.gov/content/index).

## Acknowledgements

The authors wish to thank the South African National Research Foundation (NRF) for funding this research under the South African Centre for Epidemiological Modelling and Analysis (SACEMA) bursary. SLKP and SW were supported in part by the U.S. National Institutes of Health (R01 AI134384 and AI140970, GM144468). DPM is supported by the Wellcome Trust (222574/Z/21/Z).

# Additional information

## Funding

| Funder | Grant reference number | Author |
|---|---|---|
| South African Centre for Epidemiological Modelling and Analysis | PhD Study Bursary | Rita Sianga-Mete |
| U.S. National Institutes of Health | R01 AI134384 | Sergei L Kosakovsky Pond Steven Weaver |
| U.S. National Institutes of Health | R01 AI140970 | Sergei L Kosakovsky Pond Steven Weaver |
| U.S. National Institutes of Health | GM144468 | Sergei L Kosakovsky Pond Steven Weaver |
| Wellcome Trust | 10.35802/222574 | Darren P Martin |

The funders had no role in study design, data collection and interpretation, or the decision to submit the work for publication. For the purpose of Open Access, the authors have applied a CC BY public copyright license to any Author Accepted Manuscript version arising from this submission.

## Author contributions

Rita Sianga-Mete, Conceptualization, Formal analysis, Visualization, Methodology, Writing – original draft, Project administration; Penelope Hartnady, Wimbai Caroline Mandikumba, Kayleigh Rutherford, Florence Phelanyane, Sabina Stefan, Investigation; Christopher Brian Currin, Steven Weaver, Methodology; Sergei L Kosakovsky Pond, Supervision, Methodology; Darren P Martin, Supervision, Writing - review and editing

## Author ORCIDs

Rita Sianga-Mete ⬛ https://orcid.org/0000-0001-6626-4007
Christopher Brian Currin ⬛ https://orcid.org/0000-0002-4809-5059
Sergei L Kosakovsky Pond ⬛ https://orcid.org/0000-0003-4817-4029
Darren P Martin ⬛ https://orcid.org/0000-0002-8785-0870

Reviewer #1 (Public Review): https://doi.org/10.7554/eLife.87361.3.sa1
Reviewer #2 (Public Review): https://doi.org/10.7554/eLife.87361.3.sa2
Author response https://doi.org/10.7554/eLife.87361.3.sa3

# Additional files

## Supplementary files
MDAR checklist

Supplementary file 1. Summary of the viral genome component datasets used in the study.

## Data availability

We obtained viral nucleotide sequences from the National Centre for Biotechnology Information Taxonomy database (http://www.ncbi.nlm.nih.gov/taxonomy) and the Los Alamos National Laboratory HIV sequence database (https://www.hiv.lanl.gov/content/index). These included gene and whole-genome sequences for viruses with ssRNA, ssDNA, dsRNA, and dsDNA genomes (datasets are summarized in *Supplementary file 1*).

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
