## [Editor Report · eLife Assessment]

This **valuable** study revisits the effects of substitution model selection on phylogenetics by comparing reversible and non-reversible DNA substitution models. The authors provide **solid** evidence that (1) it can be beneficial to include non-time-reversible models in addition to general time-reversible models when inferring phylogenetic trees out of simulated viral genome sequence data sets, and that (2) non time-reversible models may fit the real data better than the reversible substitution models commonly used in phylogenetics, a finding consistent with previous work.

---

## [Referee Report · Reviewer #1 (Public Review)]

The study by Sianga-Mete et al revisits the effects of substitution model selection on phylogenetics by comparing reversible and non-reversible DNA substitution models. This topic is not new, previous works already showed that non-reversible, and also covarion, substitution models can fit the real data better than the reversible substitution models commonly used in phylogenetics. In this regard, the results of the present study are not surprising.

---

## [Referee Report · Reviewer #2 (Public Review)]

The authors evaluate whether non time reversible models fit better data presenting strand-specific substitution biases than time reversible models. Specifically, the authors consider what they call NREV6 and NREV12 as candidate non time-reversible models. On the one hand, they show that AIC tends to select NREV12 more often than GTR on real virus data sets. On the other hand, they show using simulated data that NREV12 leads to inferred trees that are closer to the true generating tree when the data incorporates a certain degree of non time-reversibility. Based on these two experimental results, the authors conclude that "We show that non-reversible models such as NREV12 should be evaluated during the model selection phase of phylogenetic analyses involving viral genomic sequences". This is a valuable finding, and I agree that this is potentially good practice. However, I miss an experiment that links the two findings to support the conclusion: in particular, an experiment that solves the following question: does the best-fit model also lead to better tree topologies?

[Editors' note: the reviewers were sent the revised submission and rebuttal and based on their response, an amended eLife Assessment has been formulated.]

---

## [Author Response]

The following is the authors’ response to the original reviews

**eLife Assessment**
This valuable study revisits the effects of substitution model selection on phylogenetics by comparing reversible and non-reversible DNA substitution models. The authors provide evidence that (1) non time-reversible models sometimes perform better than general time-reversible models when inferring phylogenetic trees out of simulated viral genome sequence data sets, and that (2) non time-reversible models can fit the real data better than the reversible substitution models commonly used in phylogenetics, a finding consistent with previous work. However, the methods are incomplete in supporting the main conclusion of the manuscript, that is that non time-reversible models should be incorporated in the model selection process for these data sets.

The non-reversible models should be incorporated in the selection model process not because the significantly perform better but only because the do not perform worse than the reversible models and that true biochemical processes of nucleotide substitution does support the science of non-reversibility.

**Reviewer #1 (Public Review):**
The study by Sianga-Mete et al revisits the effects of substitution model selection on phylogenetics by comparing reversible and non-reversible DNA substitution models. This topic is not new, previous works already showed that non-reversible, and also covarion, substitution models can fit the real data better than the reversible substitution models commonly used in phylogenetics. In this regard, the results of the present study are not surprising. Specific comments are shown below.

True

It is well known that non-reversible models can fit the real data better than the commonly used reversible substitution models, see for example,
https://academic.oup.com/sysbio/article/71/5/1110/6525257

https://onlinelibrary.wiley.com/doi/10.1111/jeb.14147?af=R
The manuscript indicates that the results (better fitting of non-reversible models compared to reversible models) are surprising but I do not think so, I think the results would be surprising if the reversible models provide a better fitting.I think the introduction of the manuscript should be increased with more information about non-reversible models and the diverse previous studies that already evaluated them. Also I think the manuscript should indicate that the results are not surprising, or more clearly justify why they are surprising.

The surprise in the findings is in NREV12 performing better than NREV6 for double stranded DNA viruses as it was expected that NREV6 would perform better given the biochemical processes discussed in the introduction.

In the introduction and/or discussion I missed a discussion about the recent works on the influence of substitution model selection on phylogenetic tree reconstruction. Some works indicated that substitution model selection is not necessary for phylogenetic tree reconstruction,
https://academic.oup.com/mbe/article/37/7/2110/5810088

https://www.nature.com/articles/s41467-019-08822-w

https://academic.oup.com/mbe/article/35/9/2307/5040133
While others indicated that substitution model selection is recommended for phylogenetic tree reconstruction,
https://www.sciencedirect.com/science/article/pii/S0378111923001774

https://academic.oup.com/sysbio/article/53/2/278/1690801

https://academic.oup.com/mbe/article/33/1/255/2579471
The results of the present study seem to support this second view. I think this study could be improved by providing a discussion about this aspect, including the specific contribution of this study to that.

In our conclusion we have stated that:

The lack of available data regarding the proportions of viral life cycles during which genomes exist in single and double stranded states makes it difficult to rationally predict the situations where the use of models such as GTR, NREV6 and NREV12 might be most justified: particularly in light of the poor over-all performance of NREV6 and GTR relative to NREV12 with respect to describing mutational processes in viral genome sequence datasets. We therefore recommend case-by-case assessments of NREV12 vs NREV6 vs GTR model fit when deciding whether it is appropriate to consider the application of non-reversible models for phylogenetic inference and/or phylogenetic model-based analyses such as those intended to test for evidence of natural section or the existence of molecular clocks.

The real data was downloaded from Los Alamos HIV database. I am wondering if there were any criterion for selecting the sequences or if just all the sequences of the database for every studied virus category were analysed. Also, was any quality filter applied? How gaps and ambiguous nucleotides were considered? Notice that these aspects could affect the fitting of the models with the data.

We selected varying number of sequences of the database for every studied virus type. Using the software aliview we did quality filter by re-aligning the sequences per virus type.

How the non-reversible model and the data are compared considering the non-reversible substitution process? In particular, given an input MSA, how to know if the nucleotide substitution goes from state x to state y or from state y to state x in the real data if there is not a reference (i.e., wild type) sequence? All the sequences are mutants and one may not have a reference to identify the direction of the mutation, which is required for the non-reversible model. Maybe one could consider that the most abundant state is the wild type state but that may not be the case in reality. I think this is a main problem for the practical application of non-reversible substitution models in phylogenetics.

True

**Reviewer #1 (Recommendations for the authors):**
The reversible and non-reversible models used in this study assume that all the sites evolve under the same substitution matrix, which can be unrealistic. This aspect could be mentioned.

Done

The manuscript indicates that "a phylogenetic tree was inferred from an alignment of real sequences (Avian Leukosis virus) with an average sequence identity (API) of ~90%.". I was wondering under which substitution model that phylogenetic tree reconstruction was performed? could the use of that model bias posterior results in terms of favoring results based on such a model?

We have stated that the GTR+G model was used to reconstruct the tree. The use of the GTR+G model could yes bias the posterior results as we have stated in the paper too.

I was wondering which specific R function was used to calculate the weighted Robinson-Foulds metric. I think this should be included in the manuscript.

We stated that We used the weighted Robinson-Foulds metric (wRF; implemented in the R phangorn package (Schliep, 2011))

Despite a minority, several datasets fitted better with a reversible model than with a non-reversible model. I think that should be clearly indicated. In addition, in my opinion the AIC does not enough penalizes the number of parameters of the models and favors the non-reversible models over the reversible models, but this is only my opinion based on the definition of AIC and it is not supported. Thus, I think the comparison between phylogenetic trees reconstructed under different substitution models was a good idea (but see also my second major comment).

Noted

When comparing phylogenetic trees I was wondering if one should consider the effect of the estimation method and quality of the studied data? For example, should bootstrap values be estimated for all the ancestral nodes and only ancestral nodes with high support be evaluated in the comparison among trees?

Yes the estimation method and quality of the studied data should be considered. When using RF unlike wRF this will not matter but for weighted RF it does. When building the trees, using RaxML only high support nodes are added to the tree.

In Figure 3, I do not see (by eye) significant differences among the models. I see in the legend that the statistical evaluation was based on a t test but I am not much convinced. Maybe it is only my view. Exactly, which pairs of datasets are evaluated with the t test? Next, I would expect that the influence of the substitution model on the phylogenetic tree reconstruction is higher at large levels of nucleotide diversity because with more substitution events there is more information to see the effects of the model. However, the t test seems to show that differences are only at low levels of nucleotide diversity (and large DNR), what could be the cause of this?

The paired T-tests compares the wRF distances of the inferred tree real tree and the trees simulated using the GTR model verses the wRF distances of the inferred true tree from the trees simulated using the NREV12 model.

The reason why the influence of the NREV12 model on the tree reconstructed is not significantly higher at large levels of nucleotide diversity could be because at a certain level the DNR are simply unrealistic.

Can the user perform substitution model selection (i.e., AIC) among reversible and non-reversible substitution models with IQTREE? If yes, then doing that should be the recommendation from this study, correct?But, can DNR be estimated from a real dataset? DNR seems to be the key factor (Figure 3) for the phylogenetic analysis under a proper model.

Substitution model selection can be performed among reversible and non-reversible using both HyPhy and IQTREE. And we have recommended that model tests should be done as a first step before tree building. Estimating DNR from real datasets requires a substation rate matrix of a non-reversible.

The manuscript has many text errors (including typos and incorrect citations). For example, many citations in page 20 show "Error! Reference source not found.". I think authors should double check the manuscript before submitting. Also, some text is not formally written. For example, "G represents gamma-distributed rates", rates of what? The text should be clear for readers that are not familiar with the topic (i.e., G represents gamma-distributed substitution rates among sites). In general, I recommend a detailed revision of the whole text of the manuscript.

Done

**Reviewer #2 (Public Review):**
The authors evaluate whether non time reversible models fit better data presenting strand-specific substitution biases than time reversible models. Specifically, the authors consider what they call NREV6 and NREV12 as candidate non time-reversible models. On the one hand, they show that AIC tends to select NREV12 more often than GTR on real virus data sets. On the other hand, they show using simulated data that NREV12 leads to inferred trees that are closer to the true generating tree when the data incorporates a certain degree of non time-reversibility.Based on these two experimental results, the authors conclude that "We show that non-reversible models such as NREV12 should be evaluated during the model selection phase of phylogenetic analyses involving viral genomic sequences". This is a valuable finding, and I agree that this is potentially good practice.However, I miss an experiment that links the two findings to support the conclusion: in particular, an experiment that solves the following question: does the best-fit model also lead to better tree topologies?

By NREV12 leading to inferred trees that are closer to the true generating tree as compared to GTR, it then shows that the best-fit model in this case being NREV12 leads to better tree topologies.

On simulated data, the significance of the difference between GTR and NREV12 inferences is evaluated using a paired t test. I miss a rationale or a reference to support that a paired t test is suitable to measure the significance of the differences of the wRF distance. Also, the results show that on average NREV12 performs better than GTR, but a pairwise comparison would be more informative: for how many sequence alignments does NREV12 perform better than GTR?

We have used the popular paired t-test as it is the most widely used when comparing means values between two matched samples where the difference of each mean pair is normally distributed. And the wRF distances do match the guidelines above.

The paired t-test contains the pairwise comparison and the boxplots side by side show the pairwise wRF comparisions.

**Reviewer #2 (Recommendations for the authors):**
The authors reference Baele et al., 2010 for describing NREV6 and NREV12. I suggest using the same name used in the referenced paper: GNR-SYM and GNR respectively. Although I do not think there is a standard name for these models, I would use a previously used one.

We have built studies based on the names NREV6 and NREV12. We would like to keep the naming as standard for our studies.

GTR and NREV12 models are already described in many other papers. I do not see the need to include such an extensive description. Also, a reference should be included to the discrete Gamma rate categories [1]

We included the extensive description to enable other readers who are not super familiar with these models better understanding since we have given the models our own naming different from those used in other papers.

We have added referencing for the discrete gamma rate as recommended. (Yang, 1994)

To evaluate the exhaustiveness and correctness of the results, I would recommend publishing as supplementary material the simulated data sets or the scripts for generating the data set, the scripts or command lines for the analysis, and the versions of the software used (e.g., IQTREE). Also, to strongly support the main conclusion of the manuscript, I suggest adding to the simulations section results the RF-distances of the best-fit selected model under AIC, AICc, and BIC as well.

We can go ahead and submit all the needed datasets. The simulated data RF-Distances results are available and will be submitted. We cannot however add them to the main document as this will create very long data tables.

In some instances, it is mentioned that the selection criterion used is AIC, while in others, AIC-c is referenced. Even in the table captions, both terms are mixed. It should be made clearer which criterion is being employed, as AIC is not suitable for addressing the overparameterization of evolutionary models, given that it does not account for the sample size. A previous pre-print of this article [2] does not mention AIC-c, but also explicitly includes the formulas for AIC that do not take the sample size into account, and reports the same results as this manuscript, what indicates that AIC and not AIC-c was used here. This should be clarified. It is recommended to use AIC-c instead of AIC, especially if the sample size to model parameters ratio is low [3]. Two things may be appointed here: some authors consider tree branch lengths as model free parameters and others do not. In this paper it is not specified how the model parameters are counted. AIC tends to select more parameterized models than AIC-c, and overparameterization can lead to different tree inferences, as evidenced in Hoff et al., 2016. Therefore, it is expected that NREV12 is more frequently selected than NREV6 and GTR.In my opinion, a pairwise comparison between GTR and NREV12 performance is of great interest here, and the whiskers plots are not useful. Scatterplots would display the results better.

Boxplots are meant to offer a simplified view of the results as the paired t-tests does all of the comparisons. We shall provide the scatter plots as supplementary information so that readers can get full detailed plots as recommended.

Some references are missing.

Missing references added